# What the Lactate Shuttle Means for Sports Nutrition

**DOI:** 10.3390/nu15092178

**Published:** 2023-05-03

**Authors:** George A. Brooks

**Affiliations:** Exercise Physiology Laboratory, Department of Integrative Biology, University of California, Berkeley, CA 94720, USA; gbrooks@berkeley.edu

**Keywords:** lactate shuttle, nutrient, energy, gluconeogenic precursor, blood buffer, acid-base balance, mitochondrial reticulum, glycolysis, oxidative phosphorylation, lipolysis, DKA

## Abstract

The discovery of the lactate shuttle (LS) mechanism may have two opposite perceptions, It may mean very little, because the body normally and inexorably uses the LS mechanism. On the contrary, one may support the viewpoint that understanding the LS mechanism offers immense opportunities for understanding nutrition and metabolism in general, as well as in a sports nutrition supplementation setting. In fact, regardless of the specific form of the carbohydrate (CHO) nutrient taken, the bodily CHO energy flux is from a hexose sugar glucose or glucose polymer (glycogen and starches) to lactate with subsequent somatic tissue oxidation or storage as liver glycogen. In fact, because oxygen and lactate flow together through the circulation to sites of utilization, the bodily carbon energy flow is essentially the lactate disposal rate. Consequently, one can consume glucose or glucose polymers in various forms (glycogen, maltodextrin, potato, corn starch, and fructose or high-fructose corn syrup), and the intestinal wall, liver, integument, and active and inactive muscles make lactate which is the chief energy fuel for red skeletal muscle, heart, brain, erythrocytes, and kidneys. Therefore, if one wants to hasten the delivery of CHO energy delivery, instead of providing CHO foods, supplementation with lactate nutrient compounds can augment body energy flow.

## 1. Introduction

For nearly a century, lactate has been miscast as a metabolic waste, a cause of muscle fatigue and irritant. However, since discovery of the lactate shuttle in 1985 [1] and its subsequent elaboration [2,3,4,5,6,7,8], there has been a growing chorus on agreement of the diverse roles of lactate in biology [9,10,11,12]. The three main roles of lactate in physiology include oxidative energy source, gluconeogenic precursor, and signaling molecule [1,3,4,7,13]. Well documented in humans and other mammals is the role of lactate as a muscle [14], cardiac [6,15,16], and brain [17] fuel. Equally well documented is the role of lactate as a gluconeogenic precursor [18,19,20,21]. As a signaling molecule, a myokine, and an exerkine, lactate controls metabolism by redox signaling, covalent binding, lactylation of histones, amino acids and proteins, and other mechanisms. The mechanisms of lactate signaling have recently been recently reviewed [13]. In this paper, the potential to use dietary lactate sources to augment energy delivery during physical exercise is discussed.

## 2. Materials and Methods

### 2.1. Exercise and Cell Work

Initial and subsequent experimental results demonstrating the presence of the lactate shuttle mechanism were first and readily seen in isotope tracer studies on mammals [22,23] and humans [19,24,25,26], where the flux rates were large and lactate shuttling was obvious. Subsequently, in humans, simultaneous arterial-venous difference (a-v) and blood flow measurements to determine net and unidirectional fluxes and lactate oxidation rates were accomplished on resting and exercising human muscles, both before and after training [14], the heart during rest and exercise [15,16,27], and in brains of traumatic brain injury patients and healthy controls [17]. Subsequent studies on rodent brain tissues in culture and animals in vivo indicate lactate shuttling between astrocytes and neurons (the astrocyte-neuron lactate shuttle) [28,29] in the course of glutamatergic signaling and other processes [12].

### 2.2. The Postprandial Lactate Shuttle

Studies of postprandial dietary glucose flux and metabolism on rodent models and humans showed what has been termed as “glucose paradox” or “indirect pathway of hepatic glycogen synthesis” [30]. Studies on rodents and dogs indicated that dietary glucose released into the hepatic portal vein initially bypasses the liver and goes to the periphery where glycolysis converts glucose to lactate that is subsequently released into the central venous circulation and taken up from the arterial circulation by liver for glycogen synthesis. This, paradoxical, “indirect” pathway of hepatic glycogen synthesis is to be contrasted with the “direct” pathway in which dietary glucose from the gut is taken up and converted to liver glycogen on the first circulatory pass. In addition, although investigators have used different terminologies, earlier studies on rats and dogs were demonstrated in studies on mice [9,10].

The initial concept developed from studies on lab animals was replicated on human subjects showing both indirect and direct liver glycogen synthesis in healthy, postprandial humans. However, the balance of indirect and direct glucose conversion to liver glycogen appears to be species-related. Studies on human subjects confirmed that glycolysis is the main initial postprandial fate of glucose that accounts for approximately 66% of the overall disposal while oxidation and storage account for approximately 45% of the overall disposal. However, the majority of hepatic glycogen synthesis in postprandial humans (≈73%) was formed via the direct pathway [31]. Going forward, it should be possible to better understand how the meal size, composition, and timing influence the balance of direct and indirect liver and muscle glycogen synthesis rates in humans. Likely, data obtained using ^13^C-labeled nutrients (glucose, fructose, sucrose, and maltodextrins) and ^13^C magnetic resonance spectrometry (MRS) will be helpful [32].

While being obvious during exercise, during postprandial rest skeletal muscle is a major time period for dietary glucose consumption and conversion to lactate. Mammalian muscle is a heterogeneous tissue containing different types of muscle fibers and circulatory and connective tissue networks all with different metabolic characteristics [33,34]. Overall, mammalian muscles are glycolytic because of large capacities for glycolysis and glycogenolysis. Nonetheless, on a secondary level of classification, muscles can be classified according to oxidative capacities. As a group, red and pink muscle fibers are highly oxidative. For example, postural muscles (e.g., soleus and erector spinae) are alternatively termed intermediate (i.e., pink) or Type I fibers. In many species, deep vastus and lateral gastrocnemius muscles are bright red and termed Type IIA fibers. In contrast, white, fast twitch fibers are termed Type IIX (in humans) or IIB (in rodents). Hence, results of the above-cited studies on the glucose paradox are distinguished by results of studies on canine muscles showing greater postprandial perfusion and glucose uptake in muscles containing predominantly oxidative Type I and IIA fibers than in muscles containing predominantly Type IIB/X fibers [35,36]. Therefore, in contrast to the original lactate shuttle description, where fast twitch white, Type IIB fibers are lactate producers and drivers of cell-cell and organ-to-organ lactate shuttles, highly oxidative Type I and IIA fibers drive lactate production and flux under postprandial conditions. In addition, as already mentioned, more recently, by using isotope tracers, concentration, and blood flow measurements, these results of postprandial lactate shuttling were replicated on mice (Figure 1) [9,10].

### 2.3. Role of the Gut in Lactate Shuttling

Admittedly, results of the role of the upper and lower sections of the intestinal tract are sparse with regard to lactate formation as part of the postprandial lactate shuttle mechanism, but some data are available from studies on animal models and human subjects.

With regard to the upper bowel, the metabolic fate of a gastric glucose load given to unrestrained rats bearing a portal vein catheter has been measured. In those experiments, a porto-peripheral lactate gradient was present, reflecting the production of lactate in or by the intestine [37]. Similarly, data in support of the upper intestine glycolysis leading to lactate production in humans is scant; however, data from the sports nutrition field are helpful. In the Tappy Laboratory, combinations of ^13^C-labeled glucose and -fructose were given to human subjects to evaluate the use of oral CHO energy sources in sports drinks. The investigators observed that carbon atoms from orally ingested ^13^C-tracer fructose appeared in the systemic circulation as ^13^C-labeled lactate [38,39]. Hence, there is evidence for postprandial splanchnic lactate release in humans following the ingestion of one CHO energy source, fructose. In addition, although not yet studied, similar results might be expected from oral consumption of the disaccharide sucrose (glucose + fructose) (Figure 1).

With regard to the lower bowel and the intestinal microbiome, a gut-soma lactate shuttle has previously been suggested [4,40], but the phenomenon and its mechanisms are essentially unstudied. From nutrition science, we know that pre- and pro-biotic dietary components favorably affect gut fermentation and health [41]. In addition, data indicate relationships between microbiota and the prevalence of insulin resistance and metabolic syndrome [42]. Despites the active role of gut microbiota in fermentation and glycolysis, the concentration of lactate in human feces is relatively low (<5 mM), in part because of the presence of bacteria that convert lactate to butyrate [43]. More recently, investigators have observed the unusual presence of members of the genus Veillonella in the stools of marathon runners [44]. Unfortunately, those investigators were unaware of the importance of lactate shuttling during exercise and did not consider a scenario in which the gut supplies lactate, a fermentation product that is exported from the gut into the systemic circulation via sodium-mediated monocarboxylate transporters (sMCT) [45,46]. Future efforts to trace carbon flows following delivery of CHO nutrition to resting individuals and individuals during hard exercise and the subsequent recovery period might show that the lower bowel supports athletes’ efforts by fueling working muscles during exercise and then by clearing lactate from the circulation during the recovery period.

### 2.4. Brainless Sports Nutrition

Historically, exercise and environmental physiologists have focused on exercise performance as supported by integrated functioning of the lungs, heart, and circulatory and musculoskeletal systems [47,48,49]. In contemporary physiology, we know that the heart and red skeletal muscle are continuously fueled by lactate shuttles. However, the brain is also fueled by lactate, like the heart [6] and red skeletal muscle [5], and lactate is preferred as an energy source over glucose [12,17,50]. Recalling that all voluntary motions are neuromuscular activities, it may be necessary to evaluate the importance of fueling the brain by lactate supplementation, particularly during prolonged, strenuous and fatiguing physical endeavors. To paraphrase the sentence written by Glenn et al., “lactate is brain fuel in human traumatic brain injury and normally” [17].

### 2.5. Sour News

For the lactate shuttle science to be applied to sports nutrition, the question is “what, when and how to supply lactate nutrition?”. Should athletes be drinking lactic acid, kir, sour kraut, and kimchi juice cocktails while chewing on sourdough bread? Probably not! What then are the alternatives for safe, efficacious and generally accepted means of lactate supplementation? Some answers may stem from technology from the enteric, clinical nutrition field to support injured and critically ill patients.

Lactated Ringers’ solution has an alkalinizing effect when infused vascularly to manage metabolic acidosis. However, treatment of acidosis with lactated Ringers is far from optimal, because the solution is dilute and is often comprised of a racemic D- and L-lactate mixture, the “L” enantiomer being efficacious, while the “D” enantiomer is neuro- and cardio-toxic [51,52]. Thus, avoiding the “D” enantiomer is key, and consequently, concentrated sodium L-lactate solutions are used in cardioplegic solutions for management of heart failure and myocardial infarction [53]. For the treatment of traumatic brain injury, vascular infusion of half molar sodium L-lactate has been used [54,55,56]. In addition, even though the patient blood lactate concentration is a harbinger of poor outcome in the much feared condition of sepsis, vascular infusion of L-lactate is being considered [57,58,59]. Further, in our studies using the lactate clamp technology on healthy young resting and exercising men [60,61,62,63], the alkalinizing effect of sodium L-lactate was obvious as was the preference of lactate over glucose as a fuel energy source. Knowledge of how to use crystalloid and colloid solutions to resuscitate dehydrated, injured and ill patents is well developed, but likely, they contain inappropriate amounts of salt in the form of sodium ion and, therefore, may not be appropriate as vehicles to support carbohydrate energy needs of high-performance athletes. Importantly, for food products, there are necessary regulatory constraints as described below.

### 2.6. Why Lactate and Not Other Bioenergetic Substances?

Those experienced in studying cellular respiration and others might think of other substances that are bioenergetic at the mitochondrial level to be included in a sports drink; such substances would include pyruvate, succinate, glutamate, palmitate, and palmitoyl carnitine that are commonly used to study mitochondrial respiration. With the exception of pyruvate, the writer does not know studies using vascularly infused or orally consumed tracer-labeled alternative energy substrates in exercising humans.

Pyruvate: Carbon-tracer-labeled pyruvate has been infused to study carbohydrate metabolism, but investigators have been frustrated because infused pyruvate is rapidly converted to lactate by the enzyme LDH in erythrocytes [64] and lung parenchyma [65]. Again, to the knowledge of the writer, tracer-labeled pyruvate has not been fed to athletes or others, but via actions of intestinal mucosa and liver, the inevitable fate would likely be conversion to lactate. During rest, the lactate/pyruvate is 10 at the minimum and rises to 500 or more during moderate intensity exercise [66]. Hence, unless there is another purpose, such as carrying a cation to the GI tract, adding pyruvic acid or sodium-pyruvate to a sports drink mix would provide little advantage.

Succinate: In studies on isolated mitochondrial fragments (the cellular respiratory apparatus is organized in a reticulum or network. When “mitochondria” are isolated from muscle or other tissue, the preparation contains damaged fragments of the reticulum. Hence, the term “mitochondria” should probably be abandoned. Still, even with mitochondrial fragments, succinate donates electrons at complex 2 of the mitochondrial electron transport chain. However, little information exists on bioavailability of ingested succinate compounds by healthy persons during physical activity. Plasma succinate levels are low (50 micromolar, µM), to be compared with millimolar, mM) quantities of glucose and lactate. However, in low doses, succinate is widely used as a carrier for organic compounds such as the anti-inflammatory agent hydrocortisone, β-1 blocker metoprolol, and the antibiotic chloramphenicol. In addition, like sulfate, succinate is used to carry ferrous ion in the treatment of iron deficiency anemia. Of concern is the fact that high levels of plasma succinate are associated with obesity, diabetes, and cardiovascular disease [67]. Etiology of the relationships between elevated plasma succinate and chronic diseases is unknown, but a cellular backup and excretion of succinate into plasma due to poor mitochondrial respiratory capacity is plausible. At present, little is known to justify dietary succinate supplementation to healthy sportspeople during exercise.

Glutamate: Glutamate is another substance that donates electrons to the electron transport chain of isolated mitochondrial fragments. Neither glutamate nor glutamine are essential amino acids, but their presence is important for growth and development. Mono-sodium glutamate (MSG) has been used in some cuisines to enhance flavor and brighten vegetables; a concern related to MSG ingestion has been the side effect of headache, but that response is highly variable. The plasma concentration of glutamate is low (50–20 µM), and oral glutamate supplementation of adults, infants, and children has been studied [68]. As with glucose and lactate, intestinal glutamate absorption is sodium-ion-mediated with the sodium gradient established by a sodium-potassium ATPase in the jejunum and ileum. The plasma flux of orally consumed glutamate has not been measured in exercising humans, and the concentration and absorption of glutamate is slower and lower compared to those of dietary carbohydrates giving rise to glucose and lactate. Still, glutamate supplementation may eventually prove to be beneficial to physically active sportspeople and others.

Fatty acids, palmitate, and palmitoyl/carnitine: Even in lean athletes, body triglycerides in white adipose tissue (WAT) provide our greatest energy reserve. On a unit mass basis (e.g., gram), triglycerides provide more than twice the enthalpy (combustion energy: 9.4 Cal/g and 4.7 Cal/L O_2_) than do carbohydrates (4.2 Cal/g and 5.05 Cal/L O_2_). However, per unit of oxygen consumption, lipids yield 4.69/5.05 Cal/L O_2_ or 94% of the energy available from CHO oxidation. As given by the pulmonary exchange ratio (R or RER = VCO_2_/VO_2_), in the transition from post-absorptive rest to heavy exercise, the crossover from dependence on lipid to CHO oxidation occurs in the range of 50–65% VO_2_max [69]. Hence, compared to CHO energy sources, lipids are less preferred fuel sources with actually less lipid oxidation at high relative exercise intensities for exercise than during post-absorptive rest.

Dietary lipid supplementation during exercise is not in favor, because lipid absorption is slow. Lipid absorption occurs in the lower small intestine, with micelles and fatty acids gaining access to the systemic circulation via lymph and thoracic ducts and hepatic portal vein, respectively. In blood, fatty acids are carried by albumin, and they gain access to the intracellular matrix via plasma membrane transporters, specifically the fatty acid translocators (FAT). Those are to be contrasted with GLUT-1 and -4 for glucose and monocarboxylate transporter-1 (MCT-1) and MCT-4 for lactate. FATs are less expressed and functionally slower than GLUTs or MCTs. Further, at the mitochondrial level, uptake of activated fatty acids and carnitine derivatives is inhibited by downregulation of carnitine palmitoyltransferase-1 (CPT-1) by malonyl Co-A generated from entry of lactate into the tri-carboxylic acid cycle (TCA) [70]. In addition, CPT-2 is downregulated by lactate [71]. Hence, while isolated mitochondrial fragments can oxidize palmitate and carnitine derivatives, TCA turnover and respiration rates are lower compared to those of lactate, pyruvate, and succinate. Hence, bottles on bicycles or sidelines of sports competitions are more likely to contain dilute aqueous solutions of carbohydrate derivatives than containolive oil, no matter how more tasteful the latter might be.

Ketones: The use of ketones as a supplemental energy source in the demanding physical activities practiced by sportsmen and women was discussed above. In terms of biology, compared to carbohydrate and lipids, endogenously produced ketones are the less preferred energy sources and gluconeogenic precursors. Ketones are principally used in starvation. However, ketone supplementation during hard exercise could be another matter notwithstanding the probability that lactate would block ketone transport and mitochondrial oxidation, because plasma and mitochondrial membrane ketone transport is via MCTs. This view would change, of course, should there emerge kinetic tracer studies using a vascularly infused or orally consumed ketone showing preferential ketone over CHO oxidation during exercise. To date, limited data on vascularly infused substances into resting humans show competitive inhibition among energy substrates and specifically β-hydroxybutyrate substitutes for, and inhibits endogenous lipid, not CHO oxidation [72].

In summary on this section, it is clear that carbohydrate derivatives glucose and lactate are more readily absorbed and utilized compared to other known dietary energy sources.

### 2.7. Generally Accepted as Safe

Compared to the rules, regulations, and clinical testing requirements of pharmaceuticals, the inclusion of compounds into commercially available foods is relatively loose and unsophisticated. In the US and Europe, governmental agencies establish lists of acceptable food ingredients. In the United States, “GRAS” is an acronym for the phrase Generally Recognized As Safe. Under Sections 201(s) and 409 of the Federal Food, Drug, and Cosmetic Act, any substance that is intentionally added to food is a food additive, which is subject to premarket review and approval by Food and Drug Administration (FDA), unless the substance is generally recognized among qualified experts, as having been adequately shown to be safe under the conditions of its intended use, or unless the use of the substance is otherwise excepted from the definition of a food additive (https://www.fda.gov/food/food-ingredients-packaging/generally-recognized-safe-gras, accessed on 29 April 2023). Translation, if we consume a food ingredient for a period of time without ill effects, typically it is a GRAS food ingredient in the US. Historically in the US, the FDA has focused on the probability or likelihood of hazards or untoward things happening after the ingredient is consumed in the human population. Across the Atlantic, the European Food Safety Authority (EFSA) provides a similar scrutiny. In general, the EFSA approach is more precautionary, in that it gives attention not only to probabilities of something going wrong, but also to the mere possibility of something going wrong. Sometimes, in contrast to the use of pharmaceuticals, purported food additives are not approved, because they do not exist in nature or foods long-considered to be healthful. Hence, some food additives allowed in the US are not allowed in Europe. For example, a unique and novel double ketone supplement is available in the US, but not in Europe [73]. In this case, its utility by sportsmen and women is little effected, because ketones share plasma cell membrane lactate transporters (MCTs), but for ketone transport, the apparent KM is much higher and Vmax much is lower than for lactate [74,75]. Consequently, there is no opportunity for ketones to become a major fuel energy source during exercise, if lactate and other carbohydrate energy sources are available, either produced endogenously or provided via oral supplementation.

## 3. Discussion

To reiterate from the Introduction, since the discovery of the lactate shuttle in 1985 [1] and subsequent elaboration [2,3,4,5,6,7,8], there has been a growing chorus on agreement on the diverse roles of lactate in biology [9,10,11,12,76]. In addition, to reiterate from the Introduction, even though for nearly a century the role of lactate was miscast, science has won out. Hence, despite that early investigators completely misunderstood lactate biochemistry and physiology, basic mechanisms to produce and utilize lactate have always been in place. As reviewed briefly above, from CHO food sources, the body produces lactate for use as an energy substrate, a gluconeogenic precursor, and a signaling molecule [4,77]. Given this insight, is it possible to take advantage of new knowledge and augment what human physiology does of its own accord? The answer may be “yes”; but what, when, and how to supply lactate nutrition remain questions. The resolution to these questions will likely involve finding novel, safe and effective means of delivering CHO energy in the form of lactate compounds. At present, despite the report of Azevedo et al. showing very rapid lactate oxidation following oral ingestion and the report of Lecoultre et al. showing fructose-to-lactate conversion [38], there has been little interest in developing and certifying safe lactate supplementation compounds to enhance sports performances. Seemingly, a vast field of endeavor is in the offing.

## 4. Conclusions

Via carbohydrate nutrition, glycogenolysis, and glycolysis, contraction-derived lactate fuels cells, tissues, and organs particularly during exercise. Therefore, lactate shuttling is central to normal and exercise physiology, because during exercise muscle-derived lactate fuels red skeletal muscle [14] and the heart [6] and increases brain executive function [78]. Long regarded as a metabolic waste and a fatigue agent, contemporary recognition of the diverse roles of lactate in normal and exercise physiology, as well as in treating illnesses and injuries, is coming to the fore. The issue addressed in this article is as follows: can lactate shuttling be enhanced by providing oral nutritive support? The answer is likely “yes”. However, at present, palatable, efficacious and generally accepted as safe foods supplying carbohydrate energy in the form of lactate compounds have yet to be widely accepted in the market.

## 5. Patents

Patents relevant to the subject are shown in [79,80,81].

## Figures and Tables

**Figure 1 nutrients-15-02178-f001:**
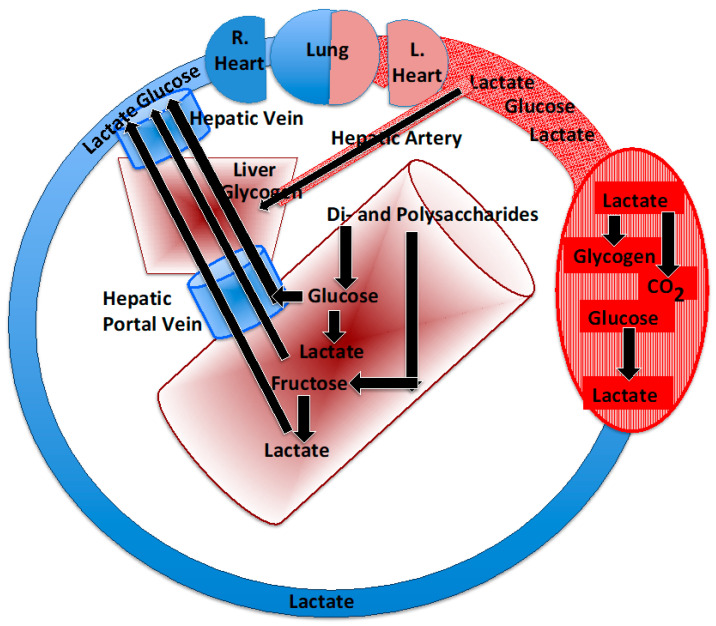
Depiction of the postprandial carbon flow depicting entry of sugars and starches into the small intestine. Polysaccharides are digested to their hexose components, mainly glucose, but also fructose. Via the hepatic portal and hepatic veins, glucose reaches the liver and then systemic circulation. In the gut lumen and intestinal wall, some glucose is converted to lactate that reaches the liver and systemic circulation via the hepatic portal and hepatic veins. Arterial fructose is low after consumption, because it is converted to lactate and glucose in the gut. After entry into the systemic circulation, glucose can be stored in muscle as glycogen or converted to lactate and oxidized or released into the systemic circulation and oxidized or converted to liver glycogen.

## Data Availability

Data are available in all papers cited.

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
