# Peer review of "What the Lactate Shuttle Means for Sports Nutrition"

_nutrients, 2023, doi:10.3390/nu15092178_

Round 1

Reviewer 1 Report

The article (opinion) "What the Lactate Shuttle Means for Sports Nutrition" by George A. Brooks is a very interesting viewpoint and update on a fundamental mechanism and its crucial importance to Sports Nutrition.

Professor Brooks has covered the Lactate Shuttle (LS) mechanism as much as possible, summarizing and correlating important data from the scientific literature, based on previous published works and his own patents!

I sincerely congratulate the author on his interesting work and the ingenious and engaging way in which he has written this scientifically sound opinion piece.

However, for the uninitiated and uninformed reader, a well-imagined diagram of the LS mechanism and pathways in the human body with an opening and consequences on sports nutrition, would be very helpful.

From this schematic picture (figure), both the journal and the readers would benefit enormously from the multiple downloads and citations it would generate, starting from the reality that:

"a picture is worth a thousand words"!

So, I recommend designing and integrating an ingenious and representative original scheme for the successful publication of this opinion!

Thank you very much!

Author Response

Reviewer #1 is thanked for their positive and constructive statements.  The suggestion to include a figure is outstanding, but will take a bit of time. 

Again, the reviews is thanked for the suggestion to include a figure.

Reviewer 2 Report

Dear Authors,

The manuscript presents a theme of significant scientific relevance.  

I suggest some figures.

Author Response

Reviewer #2 is thanked for their constructive comments. In effect, the suggestion to include a figure, or figures is accepted, and is basically the same advice as with Reviewer #1.

Again, in view of the several figures published gently, crafting a new meaningful figure may take a bit of time. 

Reviewer 3 Report

The issue is very provocative and interesting. Author described the importance of lactate in the physiology by evidencing that it has been previously treated as waste but it has changed over the years. Importantly, author has pointed out that lactate should be considered as supplementation, mainly in the context of the exercise, since it is naturally produced and used by the cells as energy source. 

However, I believe that there is an important question that was not clearly addressed: Why lactate should be supplemented ? Why not other bioenergetic intermediates ? Why not TCA intermediates, since they are naturally produced as lactate are ? For instance, succinate has been investigated as a signaling molecule produced after exercise.

However, it is important to mention that there are few studies regarding the safety, even though this seems not to be an issue for the food regulation agencies. What would be the suitable treatment by the Food regulatory agencies regarding lactate in the context of the authors' opinion ?

This reviewer consider that those are important questions to be discussed in the main text.

Author Response

Reviewer #3 is thanked for their constructive criticism, which is accepted.

Hence, in addition to a figure, or figures as suggested by Reviewer's 1 and 2, a new section titled something like "Why supplement lactate and not other bioenergetic intermediates? 

The author thanks all three reviewers for their constructive criticisms.